# A Tissue Distribution Study of Propafenone in an Intentional Fatal Poisoning Case

**DOI:** 10.3390/ijms25105202

**Published:** 2024-05-10

**Authors:** Žofia Nižnanská, Alexandra Hengerics Szabó, Marián Masár, Roman Szucs, Ján Šikuta, Ľuboš Nižnanský

**Affiliations:** 1Institute of Forensic Medicine, Faculty of Medicine, Comenius University Bratislava, Sasinková 4, 81108 Bratislava, Slovakia; szaboova66@uniba.sk; 2Department of Forensic Medicine and Toxicology, Health Care Surveillance Authority, Antolská 11, 85107 Bratislava, Slovakia; jan.sikuta@gmail.com; 3Department of Analytical Chemistry, Faculty of Natural Sciences, Comenius University Bratislava, Mlynská Dolina, Ilkovičova 6, 84215 Bratislava, Slovakia; marian.masar@uniba.sk (M.M.); roman.szucs@uniba.sk (R.S.); 4Department of Chemistry, Faculty of Education, J. Selye University, Bratislavská cesta 3322, 94501 Komárno, Slovakia; szaboszandi87@gmail.com

**Keywords:** propafenone, tissue distribution, intoxication, GC-MS/MS, suicide, poisoning, toxicology

## Abstract

Propafenone (PPF) belongs to the class 1C antiarrhythmics and can cause electrocardiogram-associated adverse/toxic effects. Cases of PPF intoxication are rarely investigated. We developed a novel and selective GC-MS/MS method for the determination of PPF and its tissue distribution in an intentional fatal poisoning case, which is applicable to PPF quantification in the range of therapeutic to lethal concentrations in complex post-mortem samples. A simple and effective sample pretreatment was applied to all analyzed samples. PPF was determined without the need for dilution, even in highly complex samples containing a wide range of analyte concentrations. Quantification was performed using the standard addition method, developed and validated according to the ICH M10 guidelines. The obtained results indicated that the PPF concentration in the serum from blood taken while alive, before therapy, was the highest ever reported in the literature. Despite the intensive therapy after the patients’ admission, the PPF concentrations in the lungs, spleen, femoral blood and cardiac blood were fatal or abnormally high. On the other hand, the concentrations in the liver and skeletal muscle were lower or approximately the same as observed in cases with therapeutic doses. To the best of our knowledge, the distribution of PPF has not been investigated in fatal intoxication cases and can be helpful in clinical or forensic toxicology.

## 1. Introduction

Propafenone (PPF) is a class 1C antiarrhythmic drug, used for irregular heartbeats, usually in atrial fibrillation. It is most often used in the treatment of supraventricular arrhythmias and ventricular tachyarrhythmias. PPF has high affinity to the voltage-gated cardiac sodium channel (Na_v_1.5), which is normally found in intercalated discs and along the lateral membranes of cardiomyocytes [1]. This sodium channel can be in a closed, open or inactivated conformational state [2]. After activation, i.e., the opening of the channel, the subsequent influx of sodium cations leads to membrane depolarization, resulting in the generation of a cardiac action potential, and cardiomyocyte excitation occurs (the initiation of the heartbeat) [3]. PPF binds with high affinity to the channel from the intracellular side in its open state and has a negative inotropic and chronotropic effect [4,5]. According to the latest conformational studies, PPF binds from the lipid bilayer through the side cleft in the sodium channel. It physically blocks the flow of sodium ions into the intracellular space [3]. In addition, PPF, unlike other class 1C antiarrhythmics (e.g., flecainide), also has a beta-blocking effect [6].

Medicines containing PPF are prescribed as immediate-release or as sustained-release tablets. PPF from a sustained-release tablet persists in the body longer and at a lower concentration, which ensures milder adverse effects. Several studies have shown that at therapeutic doses, poor metabolizers experience a significantly higher PPF concentration and elimination half-life [7]; thus, at therapeutic doses, which can become toxic, various adverse effects are observed. Zhender et al. [8] observed nausea and vomiting at short-term higher therapeutic doses and the exacerbation of ventricular extrasystoles during long-term treatment. Various studies also describe a proarrhythmic adverse effect, such as wide complex tachycardia or QRS broadening [9]; electrocardiogram-associated adverse events (prolonged QRS and QTc, first- and second-degree atrioventricular nodal block, bradycardia); and systematic adverse events (hypotension, dizziness, dysgeusia, fatigue, irritation and gastrointestinal intolerance) [10]. The coadministration of metoprolol and verapamil also increases the risk of toxicity [11,12]. The toxic effect (poisoning) of PPF occurs mainly in the case of intentional self-harm when higher doses of PPF are administered, which may or may not result in death. In these cases, symptoms such as collapse, vomiting and unresponsiveness, different levels of the Glasgow Coma Scale, a ventricular escape rhythm and intraventricular conduction, delayed bradycardia, hypotension, QRS prolongation and a corrected QT interval, generalized tonic–clonic seizures and cardiac arrest have been observed [13,14,15,16]. Fatal cases of PPF intoxication (five were intentional suicides and one was accidental) are rarer, characterized by significantly higher PPF concentrations in the blood than during therapy [17,18,19,20,21]. All studied cases had similarities in heart function failure, leading to cardiac arrest, as well as in the method of respiratory depression, ECG patterns (prolongation QT and QRS intervals), hypotension, bradycardia and other symptoms (vomiting, decreased consciousness, etc.). In all published cases, immediate-release medications were taken, and, in almost all cases, this was not accidental but intentional, i.e., probably suicidal use. The concentrations in other body fluids and tissues can be helpful in confirming poisoning (mainly the concentration in the liver), in observing the distribution of the drug in the body (its accumulation in some tissues) or in cases where post-mortem femoral blood is not available [22]. Only a few studies investigating the distribution of PPF in rats [23] and in humans [24,25] undergoing PPF therapy have been published in the literature. However, the blood permeates all organs; therefore, the concentration of each drug in this biological material is considered to be the most important. The concentration should correlate with its effect on the body. Although clinical poisonings by PPF with good therapeutic results are reported relatively frequently in the literature, fatal poisonings with PPF are far less described. This is also because fatal clinical cases resulting in death are not always autopsied, or toxicological tests are not always performed. A subsequent autopsy and toxicological analysis could undoubtedly provide better insights into the intoxication or the cause of the patient’s death [26,27].

In many publications, especially in the case of PPF poisoning, the validation parameters and necessary details that are essential for the correct execution and reproducibility of the extraction and separation are missing. Most publications [13,17,18,19,20,24,25] report on the use of liquid–liquid extraction (LLE) to isolate PPF from biological matrices (mainly from blood), but it is not apparent which analytical quantification procedure was applied. As PPF extraction is not standardized, the guidelines [28,29] recommend the use of the standard addition method (SAM) over the conventional matrix-matched calibration method (MMCM) as the preferred quantitation approach for complex matrices. Separation techniques are usually applied for quantitative analysis—most frequently, either high-performance liquid chromatography (HPLC) [13,18,19,20,24,25] or gas chromatography (GC) [17] combined with spectrometric detection (e.g., UV or diode-array detection). The published methods, however, suffer from a poor resolution as they benefit from the use of highly selective mass spectrometric (MS) detection. In addition, HPLC requires more control parameters than GC, while the reproducibility of HPLC is inferior when compared to GC [29]. Compared to HPLC, GC is also more accessible, cheaper, easy to operate and widely used. These advantages mean that it is preferred for application over other separation techniques. GC-MS instrumentation is usually available in every toxicology laboratory and it is considered the gold-standard analytical technique. In the case of low selectivity and/or sensitivity, mainly in complex biological matrices, GC can be used in combination with the MS^n^ technique, which is comprehensive and ensures very good separation of the target analytes, even at trace levels, as it exhibits an enhanced signal-to-noise ratio.

In the present study, a selective GC-MS/MS method for the determination of PPF in fluids and biological tissues using SAM quantitation was developed and validated. It was applied to a tissue distribution study of PPF in one case of intentional suicidal intoxication. In presented approach, only 1 g (mL) of biological material was needed. The overarching objective of this work was to develop a method applicable to various biological matrices with a very wide calibration range, able to capture therapeutic as well as highly lethal concentrations, without the need for sample dilutions before analysis. Draft ICH guidelines for quantitative analysis [28] and the application detailed in 2021 by Hasegawa and colleagues [30] were utilized as a reference for the method’s validation.

## 2. Case History

A man was found conscious in his prison cell during a morning check-up; he was sleepy, pale, sweaty, had communication difficulties and stated that he had taken a large number of pills during the previous night, with suicidal intent. The waste basket in the prison cell contained packages of 325 mg sustained-release tablets of the drug Rytmonorm (50 tablets out of 60 were missing). Empty packages of the medications nebivolol and melatonin, which the patient had been prescribed according to the medical records, were neither present in the waste basket nor in the prison cell. The hearing of the prison medical staff revealed that the man had been on a hunger strike for 7 days and consumed very little fluid and stated that, in the evening, he ingested approximately 30 tablets of 325 mg Rytmonorm. Two infusions of physiological saline were administered before he was taken to the hospital. In the hospital, blood was taken immediately and he also received another 750 mL PS and 500 mL gelafusine. His blood pressure was 98/68 mmHg (hypotension), his pulse was 62 beats per minute (bradycardia), his oxygen saturation was 99%, his temperature was 36.6 °C, his score on the Glasgow coma scale was 15 (full consciousness), his ECG rhythm was not sinusoidal and there was a spare ventricular rhythm with a wide QRS complex. Dark yellow urine was collected from a catheter with negative immunochemical screening for conventional drugs and pharmaceuticals. It was difficult to communicate with the patient because he had experienced an epileptic paroxysm. However, the patient mentioned again that he had swallowed many tablets on the previous day with suicidal intent. He did not have chest or abdominal pain, was breathing well and suffered from nausea. Approximately 1 h after admission, his blood pressure was 69/34 mmHg, his heart rate was 38–43, and his oxygen saturation was 96% with an oxygen flow of 6–8 L/min using an oxygen mask. It was no longer possible to communicate with the patient; he was confused, answered questions minimally, and adequate contact could not be established. His speech was dysarthric, minimal and unintelligible, and his position was passive. His skin was clean and pale without any icterus or cyanosis. His pupils were isochoric and his lips free of cyanosis. His pulse was no longer palpable on the periphery and upper limbs. A disturbance of consciousness followed, and, based on his persistent bradycardia and hypotension, temporary cardiac stimulation was initiated with the administration of isoprenaline and noradrenaline. The epileptic paroxysms persisted, which were terminated with diazepam. Despite this, the hypotension continued, so the dose of noradrenaline was increased to the resuscitation level, and dopamine and a concentrated lipid suspension were also administered. The patient was intubated under analgosedation and treated with artificial pulmonary ventilation. Despite the initiated resuscitation, electrochemical dissociation occurred, his blood pressure became immeasurable, and the patient died. In the past, the patient had been diagnosed with an atrial septal defect and an Amplatzer septal occluder was inserted.

## 3. Results

### 3.1. External Examination, Autopsy and Histology

The autopsy of a 53-year-old man with a height of 185 cm and a weight of 99 kg (BMI: 28.9) was performed 4 days after admission to the hospital. An external examination revealed his condition after cardiopulmonary resuscitation. The autopsy revealed macroscopic edema of the brain (1555 g), edema of the lungs (right: 750 g; left: 660 g), mild atherosclerosis of the coronary arteries and signs of hypertrophy of the heart (heart weight: 570 g; thickness of left ventricle wall: 1.4 cm), the enlargement of the heart cavities and the confirmation of previous heart surgery with the implantation of an atrial occluder. The autopsy further revealed mild steatosis of the liver, gallstones and venostasis with liquid dark red blood. The gastric content and gastrointestinal tract showed no signs of pills or medicinal powder (only mushy content, yellow-brown and brown masses). The microscopic histological examination confirmed (stained with H&E) the edema of the brain, pulmonary edema, dystelectasis and anthracosis of the lungs and focal steatosis in the liver. In the heart (the septum and antero-lateral wall of the left ventricle), microscopically, edema of the interstitium was found; most of the muscle fibers were narrowed, waved and enlarged and focally fragmented; the nuclei were grainy and enlarged and severely stained (with H&E); and focal perivascular insignificant fibrosis was found, with no signs of inflammation. In the kidneys, age-appropriate findings were observed, as well as autolysis in the pancreas. Based upon the external examination, the macroscopic appearance of the organs at autopsy and the microscopic examination of the tissues, no evidence of violence or no other pathological changes was found.

### 3.2. Preliminary Toxicological Screening

In the preliminary toxicological screening, femoral blood ethanol was not detected, and an acetone concentration of 0.06 g/kg was determined by headspace gas chromatography with a flame ionization detector. Other volatile substances, such as isopropanol, methanol, toluene, chloroform or xylene, were not present. A femoral blood sample was also routinely screened (from alkaline and basic extraction with diethyl ether) for the presence of commonly used psychoactive drugs, i.e., amphetamine, methamphetamine, 3,4-methylenedioxymethamphetamine, 3,4-methylenedioxyamphetamine, cannabis, opiates, benzodiazepines, fentanyl, phencyclidine, cocaine, ecstasy and barbiturates, with positive results for PPF, its metabolites and trimecaine, by thin-layer chromatography and GC-MS. Based on the record from the medical documentation, we also focused on the presence of melatonin and nebivolol, which, however, were not detected in the blood. The applied toxicological screening method was not validated, and PPF exhibited poor responses with this method. For this reason, we developed a more appropriate sample pretreatment method with proven recovery and a selective GC-MS/MS method for the determination of PPF in various biological matrices.

### 3.3. Standard Addition Method

This study aimed to develop a suitable extraction procedure and separation method for the determination of PPF, present at a wide concentration range, in diverse biological matrices. Since the GC-MS in selected ion monitoring (SIM) mode did not provide sufficient selectivity in various biological materials (see Figure 1A), a more selective GC-MS/MS method was developed (see Figure 1B). Because a large number of different interferents from complex biological matrices were extracted alongside PPF by benzene extraction, the chromatographic resolution was increased with a prolonged analysis time (30 min). This step was essential to ensure the sufficient separation of the PPF and internal standard (IS) from other interfering compounds. A high split ratio in the injector enabled the detection and quantitation of high PPF concentrations, above 300 µg/g, without detector saturation and without the need for analyte dilution.

Figure 1B demonstrates that GC-MS/MS in multiple reaction monitoring mode provided the sufficient isolation of PPF and IS from potentially interfering compounds, enabling reliable quantitation. The resolution between the PPF peak and the nearest eluting peak was <1 in GC-MS (partial resolution), but it was increased to >1.7 (baseline resolution in GC-MS/MS). The increased sensitivity of the GC-MS/MS method enabled a further increase in the split ratio, from 20:1 for GC-MS to 50:1 for GC-MS/MS, without sacrificing the signal-to-noise ratio.

As demonstrated in Table 1, the LOQ values varied from matrix to matrix, between 0.01 and 0.18 µg/g (mL), i.e., the presented method was able to quantitatively determine PPF from therapeutic (0.30 µg/mL) to lethal (very high) concentrations. Good linearity (R^2^ > 0.99) was obtained between the LOQ and the highest spiked amount in the gastric content (336 µg/mL). For each calibration point, the Qbias was calculated, as reported previously [31]. As can be seen from Table 1, the Qbias never exceeded 15% for any of the analyzed biological samples.

The extraction did not lead to a decrease in the IS signal in the blood samples (femoral blood, cardiac blood) when compared to the standard solution; however, a decrease in the IS signal could be observed in the liver, muscle and other samples (Figure 2). This was due to the higher complexity of solid samples, including the liver, when compared to blood samples. This decrease in the signal correlated with a decrease in recovery, e.g., the average recovery at different concentrations (see Section 5) of PPF was 95–113% for the blood and 56–77% for the liver, respectively. The average matrix effect was within ±15% for both the blood and the liver.

The intraday and interday repeatability were calculated as described in Section 5. Two samples were selected, with cardiac blood as the representative matrix for liquid samples and the liver as the representative matrix for solid, more complex matrices. Five data points were used to calculate both the intraday and interday repeatability and the results are summarized in Table 2. Sufficient repeatability was observed as the %RSD values did not exceed 15%, which is the recommended maximum value according to the ICH guidelines [28].

As seen from Table 3, PPF was present in all analyzed samples. The distribution of PPF was not uniform. The highest PPF concentration was found in the lungs and gastric content and the lowest in the femoral muscle. As expected, the blood serum taken during hospital admission, while the patient was alive, had the highest concentration of PPF, and it was the highest ever reported in the literature [32]. This concentration was four times higher than in the post-mortem femoral blood and two times higher than in the cardiac blood. This indicates that the origin of the blood sample is important for the interpretation of the blood content of PPF.

## 4. Discussion

The novel GC-MS/MS test procedure for the determination of PPF in biological fluids and tissues was validated and applied to samples taken from a patient when admitted to hospital and during an autopsy investigation. The results of the analysis are summarized in Table 3. The content of PPF in tissues and fluids can be relevant in the interpretation of its toxic effects. Together with clinical and autopsy findings, the determination of PPF can offer more detailed insights in elucidating the mechanism of action of PPF-related toxicity. To our knowledge, this is the first suicide case report extensively describing information about the drug administration, therapeutic interventions, autopsy results and the post-mortem distribution of PPF in the fluids and tissues. Such cases are usually not described in the literature and the diagnosis is not often combined with the determination of PPF in the blood; moreover, an autopsy is not always required.

The diagnosis of poisoning should be based on the observed effects on the human body, as well as the determination of whether the concentration of the xenobiotic in the blood is therapeutic, toxic or lethal. For a single oral therapeutic dose of PPF of 300–400 mg, the plasma concentrations ranged from 0.3 to 1.4 µg/mL [7]. At an oral dose of 150 mg every 6 h with a 4-day duration of administration, the reported steady state plasma concentration was up to 0.45 µg/mL [33]. Therapeutic and toxic concentrations of PPF from clinical case reports are more sporadic and often not detected and they aim only at observing the adverse effects. Here, however, one must also take into account co-medication as part of the therapy (pharmacokinetic interactions) before intake or with hospital therapy, ensuring a strong reduction in the PPF concentration in the body [34]. There are few clinical studies that indicate toxic PPF concentrations of 2.8 µg/mL (serum) after a dose of 150 mg 3× per day [11], 1.26 µg/mL (plasma) after an unknown dose and 9 h from patient’s admission to the clinic [13] or 2.11 µg/mL (plasma) and 10 h from admission to hospital after the ingestion of 4900 mg of PPF [14]. Similar manifestations of intoxication as in the above-mentioned studies were also observed by Gil et al. [15] at a dose of 4500 mg PPF and by Keramari et al. [16] in a suicide attempt with a dose of 2250 mg or a dose of 1800 mg PPF, where the PPF concentration in the blood, measured 6 h after intake, was 4.9 µg/mL [35]. Fatal cases of PPF intoxication are rarer, and the blood concentrations reach slightly higher values, i.e., 4.18 and 9.42 µg/mL [19]; 5.27 µg/mL [17]; 2.50 µg/mL in the serum, probably after a dose of 1500–3000 mg [20]; and 12 µg/mL in the blood, probably after a dose of 6000 mg PPF [18]. In the current case, the concentration of PPF in the serum was 17.4 µg/mL (Table 3) approximately 12 h after the ingestion of the tablets and significantly exceeded the toxic plasma concentrations observed in clinical poisonings with a successful recovery. The clinical symptoms (see case history) were also comparable to those observed in PPF poisoning [19]. In fatal cases, the PPF concentrations are usually measured in the blood. The literature does not describe the distribution of PPF in the blood (blood-to-plasma ratio) but describes its value based on the spiked blood of volunteers. The blood-to-plasma ratio for PPF is 0.7 and is defined as Cb (blood concentration of compound)/Cp (plasma concentration of compound) [36]. Thus, a serum PPF concentration of 17.4 µg/mL would represent a blood concentration of 12.2 µg/mL, which is well above the toxic blood and plasma concentrations and would be the highest lethal blood concentration reported in the literature [32]. This is in correlation with our estimated dose of 16,250 mg (based on the empty blisters found in the basket, as 50 tablets containing 325 mg of PPF were missing), which would also be the highest of all studies described so far.

The amount of PPF in the gastric content was approximately 28 mg (280 µg/g × volume of gastric content 100 mL (g)) (Table 3). Even if 28 mg represents an amount of PPF smaller than in one tablet, we would expect none detected after 12 h from the ingestion of the tablets and subsequent intensive therapy. One possible explanation could be that the PPF was ingested once as a large number of tablets or even in repeated doses sometime between the first ingestion and the time at which the patient was found. Thus, the tablets could have been distributed differently in the digestive tract, from the stomach to the intestines. The retention of the tablets in the stomach could also have been caused by the prolonged fasting of the patient (see Section 2), the consequence of which is the significantly slowed emptying of the stomach content [37]. In addition, PPF has a pKa of 9.27; therefore, it may be protonated and trapped in the stomach.

The patient was in intensive therapy for 6 h, ensuring the reduction of the PPF content in his body, which explains the lower concentrations in the blood taken after death. The concentration of PPF in the blood taken from the heart was 8.04 µg/g and the concentration in the blood taken from the femoral region was 4.03 µg/g (Table 3). These concentrations are still comparable to those found in fatal cases. The autopsy was performed 4 days after death, so it is very likely that the concentrations in the organism were affected by the process of post-mortem redistribution (PMR). It is known that the post-death blood concentrations of some drugs do not always represent the drug concentrations in the blood immediately before death due to PMR [38].

The PMR of propafenone is difficult to consider due to the overdose, resuscitation maneuvers and intensive therapy, as discussed below. A ratio of cardiac blood/peripheral blood concentrations greater than 3 indicates high PMR potential. Values of less than 1 indicate that the molecule does not exhibit PMR [38]. However, we found this ratio to be equal to 2, which is similar to the value of 2.4 observed by Dalpe et al. [39]. Resuscitation maneuvers and overdoses can distort this ratio [38], as was probably the case in our study, and it thus does not clearly indicate the PMR potential.

McIntire states that analytical data on the liver/peripheral blood ratio are relevant in examining the PMR [40]. A ratio of less than 5 indicates that the drug was subjected to post-mortem redistribution only minimally or not at all. The ratio of 1.3 observed in our study (5.31/4.03, Table 3) rather suggests the possibility that the PPF was only minimally subjected to the PMR process. However, the duration of the intensive therapy (approximately 6 h) certainly changed the concentration of PPF in the patient’s body, which complicates its interpretation in terms of PMR. G. Skopp stated that interpretative problems may arise from lengthy treatments with intravenous fluids, from devices that automatically deliver medication by the parenteral route or from transdermal patches that have been left on the body [34].

In the presented case of intoxication by PPF, the concentrations in the liver and skeletal muscle were lower or approximately the same as in cases with therapeutic doses of PPF [24,25]. This can be explained by the fact that the patient was administered a lipid emulsion during the therapy, which is associated with good recovery in PPF poisoning [41]. One of the proposed mechanisms behind the effectiveness of the lipid emulsion is the theory of lipid uptake, as the binding of the xenobiotic to the emulsion is ensured mainly by the tissue, which contains xenobiotic receptors [42]. It is probable that the administration of a lipid emulsion, along with other therapies, reduced the PPF concentrations in the liver and skeletal muscle. Gastrointestinal decontamination, activated charcoal, whole bowel irrigation, multidose activated charcoal, urine alkalization, extracorporeal removal, antidotes and ancillary support serve as useful tools for the decontamination and elimination of drugs from the body [43,44]. In six cases, treatment for PPF intoxication was successful when PPF was taken solely [14,16,35,45,46]. Early gastric lavage with activated charcoal, ionotropic drugs, mechanical cardiopulmonary devices, extracorporeal cardiopulmonary resuscitation, blood purification, aggressive supportive treatments and sodium bicarbonate were administered and seemed crucial to the successful treatment.

It is a common finding that acute intoxication leads to high concentrations in the lungs. The concentration of PPF in the lungs (309.8 µg/g) was 77 times higher and that in the spleen was seven times higher than the concentration of 4.03 µg/g in the femoral blood, and therefore also higher than the concentrations observed in fatal cases [17,19]. Despite the intensive therapy, we found relatively high concentrations in the gastric content (280 µg/g) and urine (64.8 µg/g). In the literature, we found only two documented human cases describing a PPF distribution investigation at therapeutic doses. Similar to our case, the PPF concentrations were found to be highest in the lungs, at 85.15 µg/g (a mean of six cases, ranging from 12.7 to 141.9) and that in one case was 12 µg/g [24,25], but these were still lower than the value of 309.8 µg/g in our study (Table 3). A similar pattern was observed in the distribution of PPF in rats [23]. PPF appears to be selectively retained in this tissue.

The concentration of PPF in all analyzed samples was significantly high and could be considered toxic due to the high probability that it was even higher at the time of admission to the hospital and before the commencement of the therapy. Due to the lack of literature data, it is difficult to comment on the concentration of 27.2 µg/g of PPF in the spleen (Table 3), which was the second highest among the solid tissues, after the lungs. The spleen exhibits strong blood flow and it is the most highly perfused organ; therefore, a high concentration of PPF in this tissue could be expected. Ketola and Ojanperä [22] investigated the ratios of the concentrations of several drugs in tissues to their concentrations in femoral blood. Although PPF was not among these drugs, the obtained ratio of the spleen concentration to the post-mortem femoral blood concentration of 4.25 was one of the highest ratios described. We also observed a similarly high ratio of 6.75 (it was higher only for the lungs). However, this tissue is not recommended by the mentioned authors for quantitative drug analysis as a substitute in cases where femoral blood is not available. Nonetheless, in our case, it appears, together with the concentration in the lungs, as a good indicator of intoxication, despite the intensive therapy before death.

Based on these high concentrations, together with the concentrations in the blood taken while the patient was alive (only the serum was analyzed) and in the blood taken after death, as well as on the case history and autopsy findings, the cause of death of the studied patient was PPF intoxication and the manner of death was intentional self-harm. We can explain the mechanism of death as follows. Despite the intensive therapy, the toxic influence of the sustained-released PPF (in a large number of ingested tablets), which affected the receptors in the heart muscle—with a negative inotropic and chronotropic effect—led to heart failure, with pulmonary and brain edema (swelling) and impaired consciousness, and, after exceeding the adaptation mechanisms, it led to cardiac arrest and thus directly to death.

## 5. Materials and Methods

### 5.1. Chemicals and Reagents

Benzene (99%), water (HPLC grade), sodium hydroxide (NaOH, 97%), hydrochloric acid (HCl, 37%), propafenone hydrochloride (≥98%), ephedrine (98%), methanol (HPLC grade) and N,O-Bis(trimethylsilyl)trifluoroacetamide (BSTFA) with 1% trimethylchlorosilane (TMCS) were supplied by Merck (Bratislava, Slovakia).

### 5.2. Stock and Working Solutions

Stock solutions of PPF and ephedrine were prepared at a concentration of 1 mg/mL in methanol. Working solutions of PPF were prepared by the dilution of the stock solution to 10 and 100 µg/mL in methanol, and they were used as standard addition method (SAM) spiking solutions. An ephedrine (IS) solution at 10 µg/mL was prepared from the stock solution by dilution with methanol. Then, 20 µL of the IS solution was added to each sample together with the SAM spiking solution before extraction. All stock and working solutions were stored at −20 °C until use.

### 5.3. Sample Collection and Preparation

Blood was taken from the patient while still alive at the hospital. Serum from the blood was stored at −20 °C until analysis. The autopsy was performed approximately 4 days after the patient’s death. The body was stored in a refrigerator at 4 °C until the autopsy, where femoral blood, cardiac blood, urine, gastric content, heart, lung, spleen, femoral muscle from the right lower limb, left kidney, right kidney, liver and brain samples were collected. All materials were stored at −20 °C until analysis.

Due to the diversity (e.g., liquids and solids) and complexity of the analyzed samples, the SAM was chosen for PPF quantitation. Each collected sample was divided into six equal parts, with 1 mL for liquid samples or 1 g for homogenized tissues, and placed in 20 mL closable gas-tight vials. Then, 1 mL of deionized water was added to every sample, followed by the PPF spiking solution and IS solution. The PPF concentrations in the SAM spiking solutions are shown in Table 1, and each prepared sample contained 0.20 µg/mL of IS.

The extraction of PPF from different matrices is challenging due to its physicochemical properties, e.g., its polar surface area is 58.6 Å2 (calculated using the ACD Labs software, ACD/Labs, https://www.acdlabs.com/ (accessed on 2 March 2024), Toronto, ON, Canada), indicating its hydrophobic nature, and its logP = 3.5 (3.96, calculated using the ACD Labs software), indicating higher solubility in fat-like solvents [47]. Various solvents, such as dichloromethane, toluene, diethyl ether and benzene, were used for the extraction of PPF and IS. Compared to benzene, all tested solvents showed lower extraction recovery (unpublished observation). This observation was also in agreement with the literature [23], where benzene was reported as a suitable PPF extraction solvent. Despite its toxicity, benzene was used to extract PPF from the serum, cardiac blood, femoral blood, urine, gastric content, lungs, spleen, femoral muscle and liver. None of the above-mentioned solvents were capable of extracting PPF from kidney, heart and brain tissue (unpublished observation). For this reason, the removal of proteins, followed by filtration and solid-phase extraction (SPE), was also tested. Although the SAM overcame the matrix effects and low recovery rates [30], we obtained questionably low SPE extraction recovery, e.g., an average of 8% for the spiked liver. Nevertheless, we consider it appropriate to state the following concentrations (µg/g) together with the corresponding SDs obtained from 3 repeat measurements: left kidney (6.18; SD = 0.24), right kidney (7.47; SD = 0.37), heart (0.105; SD = 0.008) and brain (7.06; SD = 0.69). We leave the assessment of the relevance of these values to the reader. The above indicates that PPF extraction from diverse samples of biological origin remains a challenge.

### 5.4. Liquid–Liquid Extraction and Derivatization

After the addition of the PPF and IS solutions, all samples were shaken for 30 min in order to homogenize the distribution of the added solutions in the matrix. Then, 200 µL of 20% NaOH and 5 mL of benzene (alkaline extraction) were added, and the mixture was mixed in a shaker at 15.17 s^−1^ for 20 min. Subsequently, the organic phase was collected and the aqueous phase re-extracted with 5 mL of benzene. After removing the benzene phase, 200 µL of concentrated HCl was added to the aqueous phase and the samples were extracted again with 2 × 5 mL of benzene under the same conditions (acidic extraction). The alkaline and acid extracts were mixed together. The extracts were evaporated under a stream of nitrogen at 40 °C, dissolved in 100 µL of the derivatization agent BSTFA with 1% TMCS and derivatized for 20 min at 85 °C.

### 5.5. GC-MS/MS Instrumentation

A GC-MS/MS system (TQ 8040, AOC 20i, Shimadzu, Kyoto, Japan) was used for analysis. After LLE, 1 µL of the derivatized sample was injected with an autosampler at a 50:1 split ratio. The inlet temperature was 290 °C. Helium was used as a carrier gas, with a total flow of 57 mL/min in the inlet, while the flow on the column was 1 mL/min. The separation was performed on a SH-Rxi–5ms capillary column with dimensions of 30 m × 0.25 mm × 0.25 µm (Shimadzu, Crossbond 5% diphenyl/95% dimethyl polysiloxane). The temperature program was set to 100 °C (initial temperature, held for 1 min) with a subsequent gradient of 10 °C/min to 290 °C, where it was held for 10 min. The interface temperature was set to 250 °C and that of the ion source to 200 °C. The solvent delay was set to 3 min and the collision energy (CE) for both monitored analytes was 15 eV. In multiple reaction monitoring (MRM) mode, ephedrine was monitored in the retention window of 4.87–14.74 min at transitions 58.1→43.1 (quantifier), 58.1→56.1 and 222.2→88.1 (qualifier). PPF was monitored in the retention window of 14.74–24.61 min at transitions 72.1→43.1 (quantifier), 72.1→41.1 and 297.2→72.2 (qualifier). The total analysis time was 30 min.

### 5.6. Method Validation

In the presented work, the standard addition method (SAM) for PPF quantitation in various biological matrices, developed and validated according to the ICH M10 guidelines [28] and Hasegawa et al. [30], was used. The most significant advantage of the SAM is that it does not require a blank matrix (surrogate matrix) for the quantification of a target compound, because the same recovery and matrix effects occur in the study samples and the calibration standards [30]. Although the SAM overcomes the matrix effects and helps to even out the recovery rates across diverse matrices, some authors suggest the investigation of the recovery in the case of lipid matrices, especially when using the LC-MS/MS method [48]. We also studied the recovery and matrix effects for selected representative samples, i.e., the blood (for fluids) and liver (for tissues). The recovery and matrix effects were calculated according to the ICH M10 guidelines [28] in six different samples of blood and the liver, which were negative for PPF content.

Using the GC-MS/MS method, the standard solutions containing 1, 2, 3, 5, and 7 µg/mL of PPF and 0.20 µg/mL of IS in the derivatization solution (BSTFA with 1% TMCS) were analyzed, and the retention times of IS and PPF were monitored. Their responses, measured as peak areas, were used to calculate the recovery. The recovery of the extraction was expressed in % for the investigated concentrations (the ratio of the area of PPF/IS in the blood or liver divided by the ratio of the area of PPF/IS in BSTFA without the presence of the matrix) × 100%. The investigated concentrations for the blood and liver were identical to those measured for the standards in the derivatization agent. Matrix effects (MEs) were investigated at the same conditions as the recovery, and the ME was expressed in % for the investigated concentrations (the ratio of the area of PPF/IS in the blood or liver added after extraction and divided by the ratio of the area of PPF/IS in BSTFA without the presence of the matrix) × 100.

The LOD and LOQ values were calculated separately for each biological matrix using the formula LOD = c_analyte_ × 3/(S/N) and LOQ = c_analyte_ × 10/(S/N) [30] from the calibration point without the addition of PPF (first calibration point).

The addition quantification accuracy, defined as Qbias%, which represents the deviation between the experimental and nominal concentrations of an analyte added to a given sample, was calculated as Qbias% = [(C + n/x + n) − 1] × 100%, as reported by Coglianese et al., for each calibration point and each biological matrix. C + n is the added analyte concentration measured in each single SAM spiking point; x + n is the theoretical concentration spiked in point n of the calibration curve [31].

The intraday repeatability was calculated from five repeat measurements of the same extracted sample, a cardiac blood or liver sample without PPF addition, on the same day. The interday repeatability was calculated from five repeat measurements using the same sample over five consecutive days. Cardiac blood and liver samples were selected as representative matrices for fluid and tissue samples, respectively.

For each matrix, six calibration points with different concentrations of PPF and the constant addition of the IS was used. The actual concentrations of PPF were obtained from the regression equation where y equals zero, x = −b/a [30], using the SAM.

## 6. Conclusions

In this work, we developed and validated a new GC-MS/MS method for the determination of PPF in the serum, blood, urine, gastric content, liver, lungs, spleen and skeletal muscle. The method is highly selective, covering a wide concentration range, without the need for sample dilution. It is applicable for PPF quantification in the range of therapeutic to lethal concentrations in complex matrices. The distribution study showed that intensive therapy can greatly impair the interpretation of the blood, liver and muscle concentrations taken post-mortem in fatal PPF poisonings. Conversely, the blood taken before therapy and the concentrations in the lungs and spleen taken after death strongly indicate the possibility of their use for the interpretation of poisoning with this substance. To our knowledge, this is the first report describing the distribution of PPF in the human body during a fatal poisoning. To confirm the trend of the distribution and the lethal concentrations in individual matrices, however, more distribution studies are necessary, especially in cases without intensive therapy. This would contribute to the better elucidation of the relationship between toxic tissue levels and the observed effects. Although this study pertains to a single subject, the obtained data on the PPF concentrations and distribution in biological material other than blood indicate, with a high probability, a case of intentional suicide.

## Figures and Tables

**Figure 1 ijms-25-05202-f001:**
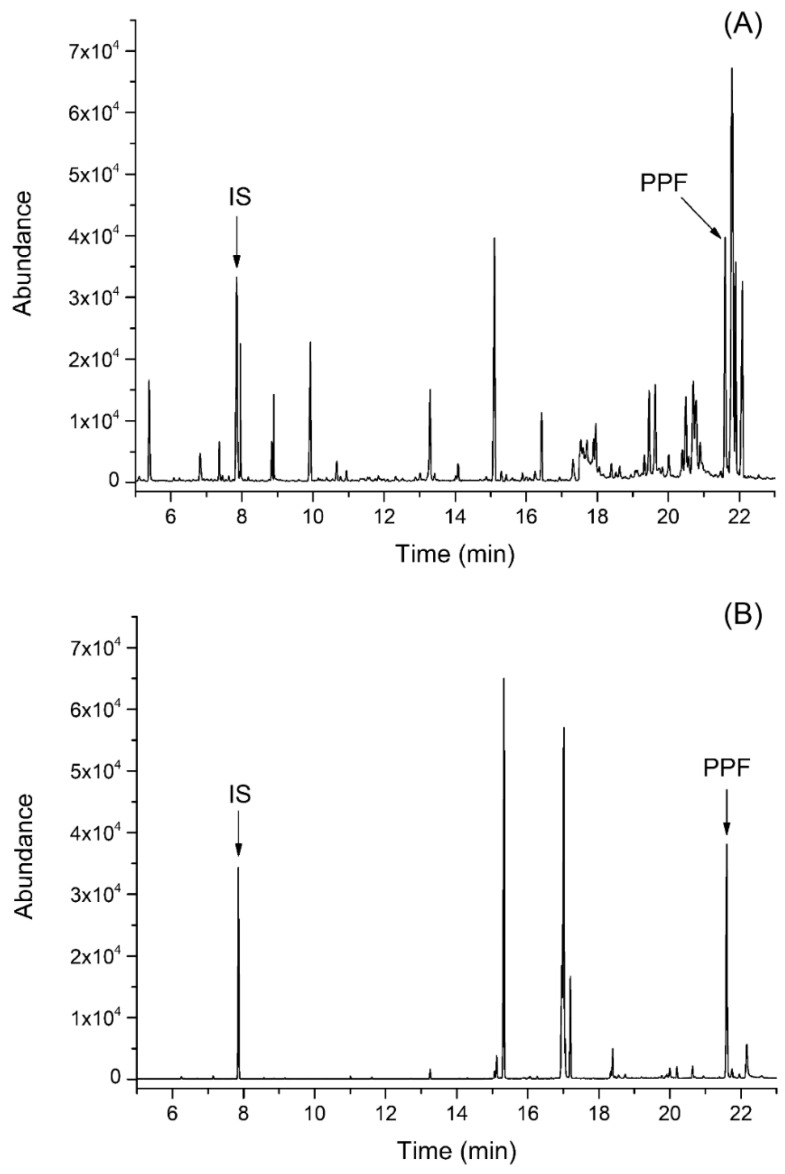
Comparison of GC-MS (trace (**A**)) and GC-MS/MS (trace (**B**)) of liver extract without addition of PPF. The GC-MS (SIM mode) signal corresponds to *m*/*z* 58 for IS and *m*/*z* 72 for PPF. The GC-MS/MS multiple reaction monitoring (MRM) signal corresponds to 58.1→43.1 for IS and 72.1→43.1 for PPF. All chromatograms correspond to derivatized samples (BSTFA with 1% TMCS).

**Figure 2 ijms-25-05202-f002:**
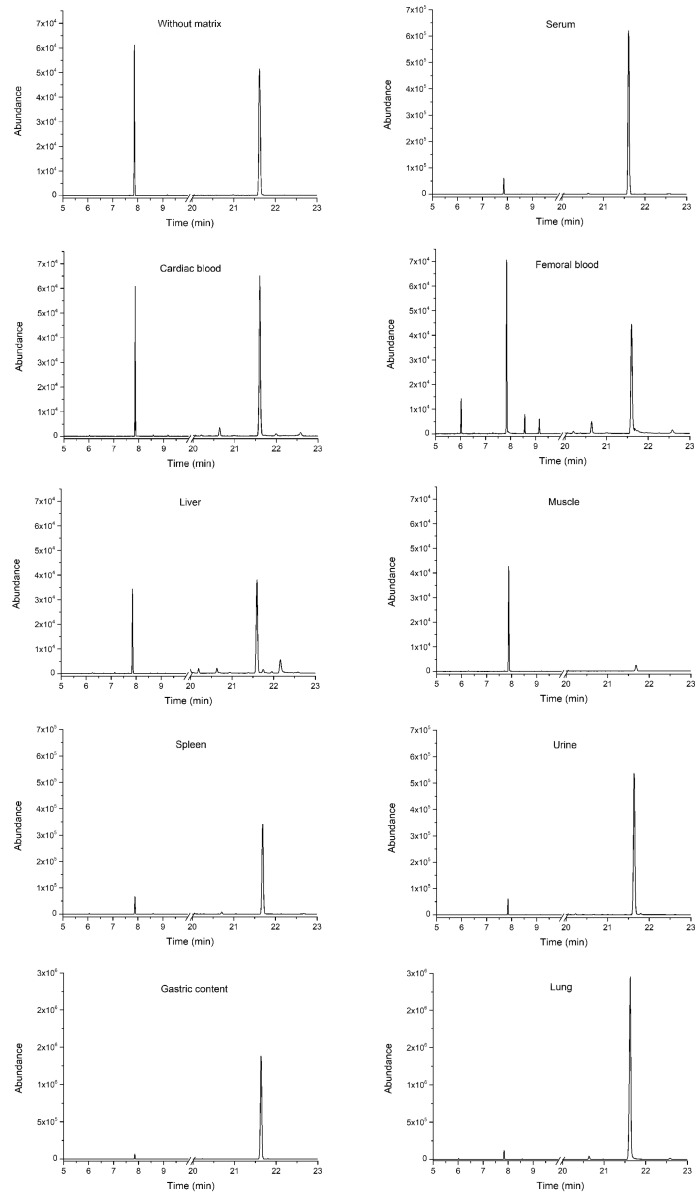
GC-MS/MS chromatograms (MRM transitions) of studied matrices without the addition of PPF (first point in the SAM calibration corresponding to real samples spiked with IS). The chromatogram labeled “without matrix” corresponds to the derivatized PPF standard at a concentration of 5 µg/mL.

**Table 1 ijms-25-05202-t001:** Calibration data, Qbias, LOD and LOQ for PPF in post-mortem body fluids (except serum) and solid tissue specimens.

Sample	PPF Addition (µg/mL; µg/g) **	Qbias (%) ***	Regression Equation (Slope; Intercept)	R^2^	LOD	LOQ
					(µg/mL; µg/g) **	
Cardiac blood	3.9–11.8	−9.9; −14.1	0.0002; 1.6	0.993	0.01	0.03
Muscle	1.8–5.2	−7.7; −8.6	0.0003; 0.1	0.995	<0.01	0.01
Lung	106–318	7.6; 14.9	0.0001; 31	0.992	0.05	0.17
Femoral blood	3.5–10.5	4.8; 8.2	0.0002; 0.8	0.998	0.01	0.03
Urine	25–75	5.3; 9.9	0.0002; 13	0.993	0.03	0.11
Serum *	8.8–26.5	−1.2; −2.1	0.0008; 14	0.995	0.05	0.18
Gastric content	112–336	3.3; 14.9	0.0001; 28	0.992	0.03	0.10
Spleen	19.1–57.4	5.9; 12.4	0.0003; 8.2	0.991	0.02	0.05
Liver	1.5–9.0	6.4; 10.8	0.0004; 2.1	0.993	0.04	0.14

* Serum taken from the patient while still alive (from blood taken before therapy); ** concentration expressed in µg/mL for liquid matrices and in µg/g for solid matrices; *** Qbias calculated for every calibration point; minimum and maximum values displayed.

**Table 2 ijms-25-05202-t002:** Intraday and interday repeatability for cardiac blood and liver matrices (*n* = 5).

Sample	Intraday Repeatability (%RSD)	Interday Repeatability (%RSD)
Cardiac blood	6.2	7.4
Liver	11.5	13.8

**Table 3 ijms-25-05202-t003:** Concentrations of PPF in post-mortem body fluids (except serum) and solid tissue specimens.

Sample	Concentration of PPF (µg/mL or µg/g) ***	RSD (%)
Cardiac blood	8.0	5.6
Femoral muscle	0.4	3.6
Lungs	309.8	1.0
Femoral blood	4.03	6.6
Urine	64.8	3.6
Serum *	17.4	5.4
Gastric content **	280.0	2.6
Spleen	27.2	4.4
Liver	5.31	11.0

The concentration of PPF corresponds to the mean value; the RSD was obtained by triplicate determination. * Serum from blood taken before therapy; ** the total volume of the gastric content was 100 mL; *** concentration expressed in µg/mL for liquid matrices and in µg/g for solid matrices.

## Data Availability

The datasets generated and/or analyzed during the current study are available from the corresponding author on reasonable request.

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
