# Peer review of "A Tissue Distribution Study of Propafenone in an Intentional Fatal Poisoning Case"

_ijms, 2024, doi:10.3390/ijms25105202_

Round 1

Reviewer 1 Report

Comments and Suggestions for Authors

The Manuscript is interesting and provides significant information in the respective field. I have some concerns about the content of the Manuscript:

-In lines 66-67, the authors should specify if fatal cases described in the literature are accidental or due to suicide manner.

- 76-79 “Although clinical poisonings by PPF with good therapeutic results are relatively abundant in the literature, fatal poisonings with PPF are far less described. This is also because fatal clinical cases resulting in death are not always autopsied, or toxicological tests are not always performed. A subsequent autopsy and toxicological analyses could undoubtedly provide better insight into intoxication or the cause of the patient's death”. Citations are needed; I suggest the following references https://doi.org/10.3390/toxics10110654 

https://doi.org/10.1191/0960327104ht454oa 

-What about autopsy findings and the exact cause of death in fatal cases in literature? Only cardiac arrest or arrhythmic deaths have been described? The authors should clarify.

- Lines 149-162. Histological heart findings should be better described: no hypertrophy? What about coronary arteries? No signs of fibrosis? The authors should clarify what they mean by atherosclerosis of the heart. What about gastric content appearance? Are there any signs of pulmonary edema?

- Lines 331-333. The authors should expand the discussion section  with literature evidence on the influence of intensive and infusion therapy on toxicological analysis results.

-The authors should clarify the exact cause of death and discuss the physiopathology in this case according to autopsy and toxicological findings.

Comments on the Quality of English Language

Minor english style and language check is needed

Author Response

We wish to express our gratitude to reviewers for the time dedicated to the review and for useful suggestions which are improving the quality and clarity of submitted manuscript.

Reviewer 1: The Manuscript is interesting and provides significant information in the respective field. I have some concerns about the content of the Manuscript:

-In lines 66-67, the authors should specify if fatal cases described in the literature are accidental or due to suicide manner.

Authors: We specified manner of dead. It was highlighted in revised manuscript.

Fatal cases of PPF intoxication (five were intentional suicides and one was accidental) are rarer, characterized by significantly higher PPF concentrations in the blood than during the therapy.

Reviewer 1:- 76-79 “Although clinical poisonings by PPF with good therapeutic results are relatively abundant in the literature, fatal poisonings with PPF are far less described. This is also because fatal clinical cases resulting in death are not always autopsied, or toxicological tests are not always performed. A subsequent autopsy and toxicological analyses could undoubtedly provide better insight into intoxication or the cause of the patient's death”. Citations are needed; I suggest the following references 

https://doi.org/10.3390/toxics10110654 

https://doi.org/10.1191/0960327104ht454oa

Authors: The above mentioned references were added to the manuscript and highlighted in revised manuscript.

Reviewer 1:-What about autopsy findings and the exact cause of death in fatal cases in literature? Only cardiac arrest or arrhythmic deaths have been described? The authors should clarify.

Authors: It was included in the introduction and highlighted in revised manuscript.

All scoped cases had similarities in heart function failure leading to cardiac arrest, way of respiratory depression, ECG patterns (prolongation QT and QRS intervals), hypotension, bradycardia and other symptoms (vomiting, decreased consciousness, etc.). In all published cases, the immediate release medicaments were taken and in almost all cases this was not accidental, but intentional, very probably suicidal use.

Reviewer 1:- Lines 149-162. Histological heart findings should be better described: no hypertrophy? What about coronary arteries? No signs of fibrosis? The authors should clarify what they mean by atherosclerosis of the heart. What about gastric content appearance? Are there any signs of pulmonary edema?

Authors: It was added to the introduction and highlighted in revised manuscript.

…..mild atherosclerosis of coronary arteries and signs of hypertrophy of the heart (heart weight 570 grams, the thickness of the left ventricle wall 1.4 cm), enlargement of the heart cavities and confirmed state after heart surgery with the implantation of an atrial occlude in the past. Autopsy further revealed mild steatosis of the liver, gallstones and venostasis with liquid dark red blood. Gastric content and gastrointestinal tract with no signs of pills, or medicinal powder (only mushy content, yellow-brown and brown masses). Microscopic histological examination confirmed (stained with H&E) edema of the brain, pulmonary edema, dystelectasis and anthracosis of the lungs, confirmed focal steatosis in the liver. In the heart (septum and antero-lateral wall of the left ventricle), microscopically, edema of interstitium was found, most of the muscle fibers were narrowed waved and enlarged, focally fragmented, nuclei were grainy enlarged, worsen stained (with H&E), focal perivascular insignificant fibrosis was found, with no signs of inflammation. In kidneys age-appropriate findings were observed as well as autolysis in the pancreas. Based upon the external examination, the macroscopic appearance of the organs at autopsy and the microscopic examination of the tissues, no evidence of violence and no other pathological changes were found.

Reviewer 1:- Lines 331-333. The authors should expand the discussion section  with literature evidence on the influence of intensive and infusion therapy on toxicological analysis results.

Authors: It was added to the discussion and highlighted in revised manuscript.

  1. Skopp described that interpretative problems may arise from lengthy treatment with intravenous fluids, from devices which automatically deliver medication by the parenteral route, or from transdermal patches that have been left on the body (G. Sopp 2010).

This can be explained by the fact that the patient was administered a lipid emulsion during the therapy, which shows good recovery in PPF poisoning [36]. One of the proposed mechanisms of the effectiveness of the lipid emulsion is the theory of lipid uptake, when the binding of the xenobiotic to the emulsion is ensured mainly from the tissue that contains xenobiotic receptors [37]. It is probable that the administration of lipid emulsion along with other therapies reduced PPF concentrations in liver and skeletal muscle. Gastrointestinal decontamination, activated charcoal, whole bowel irrigation, multidose activated charcoal, urine alkalization, extracorporeal removal, antidotes and ancillary support serve as useful decontaminations and eliminations of drugs from the body (Chandran 2019, Megarbane 2020). In 6 cases, the treatment of PPF intoxication was successful when PPF taken solely, or in combination with other medicaments (e.g., diazepam or metoprolol e.g. Ling 2018; Marano, 2018; Keramari 2021; Chu, 2020; Li, 2020)). Early gastric lavage with activated charcoal, ionotropic drugs, mechanical cardiopulmonary device, extracorporeal cardiopulmonary resuscitation, blood purification, aggressive supportive treatments, sodium bicarbonate was administered and seemed crucial in successful treatment.

Reviewer 1:-The authors should clarify the exact cause of death and discuss the physiopathology in this case according to autopsy and toxicological findings.

Authors: It was added to the discussion and highlighted in revised manuscript.

Despite the intensive therapy, toxic influence of sustained released PPF (large number of ingested tablets) which affected receptors in heart muscle - negative inotropic and chronotropic effect, led to heart failing function, with pulmonary and brain edema (swelling), impaired consciousness, and after exceeding the adaptation mechanisms, it led to cardiac arrest and thus directly to death.

Reviewer 1: Minor english style and language check is needed

Authors: English language was checked and corrections were marked by tracked changes

Reviewer 2 Report

Comments and Suggestions for Authors

Propafenone (PPF) belongs to a class 1C antiarrhythmics and can cause electrocardiogram-associated adverse/toxic effects. Cases of PPF intoxications are rarely investigated. Authors  developed a novel and selective GC-MS/MS method for the determination of PPF and its tissue distribution in an intentional fatal poisoning case, which is applicable for PPF quantification in the range of therapeutic to lethal concentrations in the complex post-mortem samples.

Quantification was performed using the standard addition method developed and validated according to the ICH M10 guidelines. As Authors  affirmed that "To the best of our knowledge, the distribution of PPF has not been investigated in fatal intoxication cases and can be helpful in clinical or forensic toxicology". 

I would suggest to reader, consider  and cite appropriately:  Disposition of toxic drugs and chemicals in man: Randall C. Baselt. Twelfth edition. Biomedical Publications Seal Beach, CA. 2020 (pages 1801-1803) .

For your convenience I attached relatives pages (1801-1803) of the book.

With regard to key words Authors indicated ( propafenone; tissue distribution; intoxication; GC-MS/MS; suicide; poisoning),  I would suggest also include: toxicology

The obtained results indicate that the PPF concentration in serum from blood, taken alive before therapy, was the highest ever reported in the literature (as above, please consider and cite the book). 

Comments on the Quality of English Language

Minor revision are required.

Author Response

We wish to express our gratitude to reviewers for the time dedicated to the review and for useful suggestions which are improving the quality and clarity of submitted manuscript.

Reviewer 2

Propafenone (PPF) belongs to a class 1C antiarrhythmics and can cause electrocardiogram-associated adverse/toxic effects. Cases of PPF intoxications are rarely investigated. Authors  developed a novel and selective GC-MS/MS method for the determination of PPF and its tissue distribution in an intentional fatal poisoning case, which is applicable for PPF quantification in the range of therapeutic to lethal concentrations in the complex post-mortem samples.

Quantification was performed using the standard addition method developed and validated according to the ICH M10 guidelines. As Authors  affirmed that "To the best of our knowledge, the distribution of PPF has not been investigated in fatal intoxication cases and can be helpful in clinical or forensic toxicology".

I would suggest to reader, consider  and cite appropriately:  Disposition of toxic drugs and chemicals in man: Randall C. Baselt. Twelfth edition. Biomedical Publications Seal Beach, CA. 2020 (pages 1801-1803) .

Authors: Citation was added to the manuscript and highlighted in revised manuscript.

Reviewer 2 :For your convenience I attached relatives pages (1801-1803) of the book.

With regard to key words Authors indicated ( propafenone; tissue distribution; intoxication; GC-MS/MS; suicide; poisoning),  I would suggest also include: toxicology

Authors: Word toxicology was added to keywords and highlighted. We would like to thank you for  attached pages.

Reviewer 2:The obtained results indicate that the PPF concentration in serum from blood, taken alive before therapy, was the highest ever reported in the literature (as above, please consider and cite the book).

Authors: Citation was added to the manuscript and highlighted in revised manuscript.

Round 2

Reviewer 1 Report

Comments and Suggestions for Authors

The authors responded to all the addressed issues. The manuscrip can be accepted in present form.

Reviewer 2 Report

Comments and Suggestions for Authors

Authors answered to all suggestions and criticism; the revised version of the paper is worthy  of consideration for the readers.